# Role of Advanced Glycation End Products on Aortic Calcification in Patients with Type 2 Diabetes Mellitus

**DOI:** 10.3390/jcm9061751

**Published:** 2020-06-05

**Authors:** Pilar Sanchis, Rosmeri Rivera, Regina Fortuny, Carlos Río, Miguel Mas-Gelabert, Marta Gonzalez-Freire, Felix Grases, Luis Masmiquel

**Affiliations:** 1Vascular and Metabolic Diseases Research Group, Endocrinology Department, Son Llàtzer University Hospital, Health Research Institute of the Balearic Islands [IUNICS-IdISBa], 07198 Palma of Mallorca, Spain; rosmeri_ri@hotmail.com (R.R.); rfortuny@hsll.es (R.F.); marta.gonzalezfreire@ssib.es (M.G.-F.); 2Laboratory of Renal Lithiasis Research, Deptartment of Chemistry, University of Balearic Islands, Health Research Institute of the Balearic Islands [IUNICS-IdISBa], 07122 Palma of Mallorca, Spain; fgrases@uib.es; 3The Spanish Biomedical Research Centre in Physiopathology of Obesity and Nutrition (CIBERObn), Instituto de Salud Carlos III, 28046 Madrid, Spain; 4Laboratory Department, Son Llàtzer University Hospital, 07198 Palma of Mallorca, Spain; 5Proteomics department, Health Research Institute of the Balearic Islands (IdISBa), 07120 Palma of Mallorca, Spain; carlos.rio@ssib.es; 6Radiology Department, Son Llàtzer University Hospital, 07198 Palma of Mallorca, Spain; mmas1@hsll.es

**Keywords:** AGEs, aortic calcification, type 2 diabetes mellitus, diabetes-related complications

## Abstract

The aim of this study was to evaluate the relationship between serum levels of advanced glycation end products (AGEs) and abdominal aortic calcification (AAC) in patients with type 2 diabetes mellitus (DM2). This was a prospective cross-sectional study. One-hundred and four consecutive patients with DM2 were given lateral lumbar X-rays in order to quantify abdominal aortic calcification (AAC). Circulating levels of AGEs and classical cardiovascular risk factors were determined. Clinical history was also registered. Patients with higher AGEs values had higher grades of aortic calcification and higher numbers of diabetic-related complications. Multivariate logistic regression analysis showed that being older, male and having high levels of AGEs and triglycerides were the independent risk factors associated to moderate-severe AAC when compared to no-mild AAC. Our results suggest that AGEs plays a role in the pathogenesis of aortic calcifications. In addition, the measurement of AGEs levels may be useful for assessing the severity of AAC in the setting of diabetic complications.

## 1. Introduction

Cardiovascular (CV) disease is the leading cause of death in patients with type 2 diabetes mellitus (DM2) in Western countries and represents a major burden on morbidity, quality of life and health resources. Thus, the identification of patients with DM2 at a high risk of a CV event is important to develop early interventions and prevention strategies. In clinical practice, patients with worse lipid profiles, higher blood pressure, obesity, hyperglycemia, cardiac hypertrophy or dysfunction, nephropathy, neuropathy and/or retinopathy tend to have a higher prevalence of CVD [1]. In recent years, however, several studies have suggested that abdominal aortic calcification (AAC) increases the risk of major CV events in patients with DM2 [2,3,4]. AAC has been related to severe coronary artery calcification and cardiovascular outcomes specially in patients with DM2 and chronic kidney disease [5,6]. Also, it has been reported that AAC predicts major CV events after an acute coronary syndrome and predicts health care costs in older men independent of CV disease status as well [7,8]. Hence, detection and scoring AAC could improve the clinical identification of patients with DM2 at risk of developing major CV events.

AAC is described as the hardening of the medial layer of the artery through deposition of hydroxyapatite crystals into the extracellular matrix [9,10,11]. Calcification occurs, amongst other factors, as a consequence of aging [12], but it can be accelerated by other diseases like renal insufficiency and diabetes [13,14]. Although several factors could accelerate or arrest the natural process of vascular calcification, the pathogenesis of vascular calcification in diabetes is not completely understood. Among the list of inducers, we can find hyperphosphatemia, vitamin D, lipids or inflammatory cytokines, whereas fetuin-A, phytate, pyrophosphate, vitamin K, osteopontin or matrix Gla protein have been described to be inhibitors of vascular calcification [15]. Consequently, further understanding of the mechanisms by which diabetes might favor this process needs to be elucidated. Indeed, high glucose appears to have an important role in the initiation and progression of this complication.

Ideally, AAC can be quantified with computed tomography (CT), lumbar X-ray and dual-energy X-ray absorptiometry (DXA) scans [16]; however, these imaging studies are not routinely performed in the clinic. Therefore, circulating biomarkers could be useful to identify those patients more prone to AAC and to develop new treatments. In this regard, low-density lipoproteins (LDL), high-density lipoproteins (HDL), glycated hemoglobin (HbA1c), osteocalcin, phosphate and advanced glycation end products (AGEs) have been identified as potential markers of AAC but literature is ambiguous and further research is needed [2].

AGEs are the irreversible products of the reaction between amino groups of proteins and sugar resulting from long-term hyperglycemia [17]. AGEs are formed at an accelerate rate under non-controlled diabetic state and play an important role in the development and progression of vascular complications in diabetes [18]. AGEs can affect cell function via intracellular glycation of proteins [19], provoke arterial stiffness by cross-linking of extracellular matrix proteins in arteries [20], and induce receptor-mediated cell activation by binding to the receptor of AGEs (RAGE) [21]. AGEs are believed to be involved in the onset and progression of atherosclerosis through multiple mechanisms and several studies indicate that AGEs and their receptors (RAGEs) play an important role in vascular calcification [22]. In individuals with chronic kidney disease (CKD), AGEs have been associated with arterial calcification [23,24]. Recently, several studies using different models of vascular smooth muscle cells (VSMC) demonstrated the implication of AGEs in calcification [25,26,27,28]. Indeed, it was shown that exposure of VSMC to AGEs induced cell calcification [29]. However large studies investigating plasma AGEs levels and their relation with AAC are scarce.

Therefore, the main objectives of the present study are: (1) to examine the association between serum levels of AGEs with AAC, quantified on lumbar X-ray; and (2) to determine the most important CV risk factors and diabetic complications associated to moderate-severe AAC in T2DM. 

## 2. Experimental Section

### 2.1. Subjects and Methods

One-hundred and four patients with DM2 were recruited consecutively at the Outpatient Diabetic Clinic of Son Llàtzer University Hospital (Palma de Mallorca, Spain). The demographic characteristics of the patients according to the AGEs levels are shown in Table 1. Briefly, patients were included if they were older than 40 years, had DM2 duration longer than 1 year, and had had a lumbar X-ray performed in the past 6 months. Exclusion criteria were: life expectancy shorter than 1 year, immunodeficiency, human immunodeficiency virus (HIV) infection, drug abuse or alcohol intake higher than 50 g/day, body mass index (BMI) higher than 40 Kg/m^2^, participation in a clinical trial in the past year, inability to attend programmed visits, illiteracy, chronic or acute infection in the past 3 months, neoplasia in the past 5 years, vasculitis, resistant hypertension, moderate or severe hepatic insufficiency or glomerular filtration rate (GFR) lower than 30 mL/min/1.73 m^2^.

The study protocol was approved by the Ethics Committee of the Balearic Islands (approval number: CEIC-IB 2523/15PI). Written informed consent was obtained from all study participants.

### 2.2. Variables Outcomes

The main outcome measures were serum level of AGEs and AAC scores. Clinical histories were obtained from the electronic medical records. Furthermore, data from anamnesis, laboratory analysis, and physical examinations were prospectively collected during the study. Physical and anthropometric measurements were determined by qualified personnel. Blood samples were collected in the morning (after 12 h of fasting). The samples were left to stand for 30 min at room temperature, and the serum was then separated by centrifugation. Biochemical analyses were performed in an automated analyzer (Cell-Dyn Sapphire and Architect ci16200, Abbott, IL, USA). Insulin was analyzed by chemiluminescent-immunometric assay (Advia Centaur, Siemens, NY, USA). Highly sensitive C-reactive protein (hs-CRP) and lipoprotein (a) (Lp [a]), were analyzed by nephelometry (Immage 8000, Beckman Coulter Inc, CA, USA). All samples were run in duplicate, and the coefficients of intra- and inter-assay variation were below 10%.

Blood pressure was measured 3 times consecutively after 5 min of rest while the subject was sitting quietly. The average of the second and third measurements was recorded. Patients using anti-hypertensive drugs and those with systolic blood pressure of 140 mmHg or more and/or diastolic blood pressure of 90 mmHg or more were categorized as having hypertension [30]. Atherosclerosis was diagnosed by having clinically significant carotid and/or femoral plaque burden documented with arterial ultrasonography. Chronic kidney disease (CKD) was diagnosed based on the estimated GFR (stage 2 CKD: 89–60 mL/min/1.73 m^2^; stage 3a CKD: 59–45 mL/min/1.73 m^2^; stage 3b: 44–30 mL/min/1.73 m^2^), calculated as previously described [31].

### 2.3. Serum Advanced Glycation End Products (AGEs) Determination

AGEs in serum samples were measured using the OxiSelect™ AGE Competitive enzyme-linked immunosorbent assay (ELISA) Kit (Cell Biolabs Inc, CA, USA), which provides rapid detection and quantification of AGE protein adducts. Quantitation was determined by comparing absorbance with that of a known AGE–bovine serum albumin (BSA) standard curve. First, an AGE conjugate was coated on an ELISA plate. The unknown AGE samples or AGE–BSA standards were added to the AGE conjugate pre-absorbed ELISA plate. After a brief incubation, an anti-AGE polyclonal antibody was added, followed by horseradish peroxidase-conjugated secondary antibody. The content of AGE protein adducts in unknown samples was determined by comparison with a pre-determined AGE–BSA standard. The intra-assay coefficient of variation was 6.9% (three replicates for each sample on the same day). The inter-assay coefficient of variation was 9.2% (3 days at the same time).

### 2.4. Lateral Lumbar Radiography of Abdominal Aorta

Lateral lumbar X-rays were performed while patients were standing, using standard radiographic equipment. A minimum of 8 cm of tissue anterior to the lumbar spine, including the abdominal aorta, was visible. The focus-film distance was 100 cm, the tube potential was 94 kV, the tube-current-time product was 33–200 mAs, and the estimated radiation dose was approximately 15 mGy.

ACC was assessed using a previously validated 24-point scale [32,33]. For this 24-point scale, calcified deposits along the anterior and posterior longitudinal walls of the abdominal aorta, adjacent to each lumbar vertebra (L1 to L4), were assessed using the midpoint of the intervertebral space above and below the vertebrae as the boundaries. Calcifications were graded as 0 (no aortic deposits), 1 (small scattered deposits less than one-third the length of the vertebral length), 2 (intermediate quantity of deposits, about one-third or more, but less than two-thirds of the vertebral length), or 3 (extensive deposits of two-thirds or more of the corresponding vertebral length). The scores were determined separately for the anterior and posterior walls; the range was 0 to 6 for each vertebral level, and 0 to 24 for the total.

All subjects were assessed independently by two graders who were blinded to patient data. To validate the X-ray assessment, double readings were performed in all patients demonstrating an excellent inter-observer agreement (intra-class coefficient of correlation R = 0.948, *p* < 0.0001).

### 2.5. Statistical Analysis

Data are presented as means and standard deviations, medians and interquartile ranges, or numbers and percentages. Patients were divided in three groups according to tertiles of serum AGEs levels: low (L < 6.5 U/mL), intermediate (I: 6.5–10.4 U/mL) and high (H > 10.4 U/mL). Intergroup comparisons of serum levels of AGEs employed one-way analysis of variance (ANOVA) and the independent-samples *t*-test (as post-hoc test); or Kruskal–Wallis test and Mann–Whitney U test (as a post-hoc test) for continuous variables. A chi-square test and Fisher’s exact test were used for categorical variables. For the second objective, patients were divided in two groups according the median of AAC scores: no-mild (AAC < 6) and moderate–severe (AAC ≥ 6). Intergroup comparisons of aortic abdominal calcification (AAC) scores employed the independent-samples *t*-test or the Mann–Whitney U test for continuous variables, and the chi-square test or Fisher’s exact test for categorical variables. Receiver operating characteristic (ROC) curves of quantitative risk factors associated to moderate–severe AAC were performed. The optimal cutoff values were determined by the maximum Youden index (J), defined as sensitivity + specificity −1. Binary logistic regression models were used to identify risk factors associated to moderate–severe AAC (AAC ≥ 6), with an AAC less than 6 (no-mild AAC) as the reference (odds ratio [OR] = 1). Analysis was performed using the stepwise backward method. A two-tailed *p*-value less than 0.05 was considered statistically significant. Statistical analyses were performed using SPSS 23.0 (SPSS Inc., Chicago, IL, USA).

## 3. Results

### 3.1. Baseline Patient Characteristics

Demographic and clinical characteristics of the three groups are shown in Table 1. Patients with higher AGEs levels were older (L: 67.5 ± 7.3; I: 66.9 ± 10.4; H: 73.7 ± 7.9 years; *p* < 0.001) and the percentage of males was also higher (L: 34.3%; I: 50.0% and H: 60.0%; *p* = 0.095), although it did not reach statistical significance. Patients with higher AGEs levels presented more diabetic complications, such as diabetic nephropathy, atherosclerosis and chronic kidney disease (*p* < 0.05). Furthermore, the number of diabetic complications increased with AGEs levels (Figure 1). The percentage of patients with three or more DM complications was 6%, 21% and 34% for low, intermediate and high AGEs groups, respectively (*p* = 0.009) while the percentage of patients with no DM2 complications were 57%, 38% and 23%, respectively. Regarding medication, a higher percentage of patients with high AGEs levels were taking calcium antagonists and antiplatelets when compared to patients with low AGEs levels (*p* = 0.025 and *p* = 0.001, respectively).

### 3.2. Laboratory Analysis Parameters

Table 2 shows the clinical and biochemical characteristics for AGEs groups. As can be seen, GFR was lower for high AGEs group compared to low and intermediate AGEs groups [L:77.8(55.7–88.9); I:83.1(50.1–98.2); H: 65.4(46.0–79.1) mL/min/1.73 m^2^; *p* = 0.040] whereas blood levels of hemoglobin, urea, creatinine, urate, albumin: creatinine ratio and glycated hemoglobin significantly increase with AGEs levels (*p* < 0.05).

### 3.3. Abdominal Aortic Calcification and AGEs Levels

There was a trend toward more AAC as serum AGEs levels increased (Figure 2). The median (inter-quartile range, IQR) of AAC 24-points score was 2 (0–5), 9.5 (4–13) and 8 (2–13) for low, intermediate and high AGEs groups, respectively (Figure 2a, *p* = 0.025). There were 23%, 68% and 63% of patients with moderate-severe AAC in the low, intermediate and high groups respectively (Figure 2b, *p* < 0.001). Furthermore, 21 (20.2%) of all patients presented no AAC and of these, 10 (47.6%), 5 (23.8%) and 6 (28.6%) presented low, intermediate and high AGEs levels, respectively.

### 3.4. Cardiovascular (CV) Risk Factors Associated with Abdominal Aortic Calcification (AAC)

Table 3 shows the clinical characteristics and laboratory parameters between patients with no-mild AAC (AAC < 6) and moderate-severe AAC (AAC ≥ 6). The group with moderate-severe AAC was older, had a higher percentage of males, more DM2 complications, more prevalence of atherosclerosis, higher percentage of smokers or ex-smokers, lower GFR and higher levels of urinary albumin, albumin: creatinine ratio, creatinine, triglycerides, HbA1c and AGEs compared to those with no-mild AAC.

ROC curves and optimal cut off values were calculated for quantitative risk factors associated to moderate-severe AAC (Figure 3). As can be seen, circulating serum AGEs upper than 7 U/mL had a sensitivity and specificity of 81.1% and 56.9%, respectively. The overall accuracy was 68.3% for AGEs above 7 U/mL and it was bigger than those for the rest of the analyzed variables (62.5% for age above 70 years; 61.5% for triglycerides upper than 146 mg/dL; and 64.4% for GFR lower than 74 mL/min/1.73 m^2^).

Univariate and multivariate logistic regression analysis were used to investigate independent factors associated to the presence of moderate–severe AAC (vs. no-mild AAC). All previously listed factors (*p* < 0.05 in Table 3) were included initially in the model before stepwise and backward elimination. The final model included age older than 70 years, being male, circulating AGEs levels upper than 7 U/mL, and triglycerides levels upper than 146 mg/dL as the independent risk factors associated to moderate–severe AAC in DM2 (Figure 4).

## 4. Discussion

Our study reports that serum levels of a pool of circulating AGEs determined by ELISA positively correlated with AAC (based on lateral lumbar X–rays) in subjects with DM2. Previously, Saremi et al. prospectively demonstrated in the *VA Diabetes Trial and Follow-up Study* that baseline plasma levels of Nε-carboxyethyl lysine (CEL) determined by liquid chromatography-mass spectrometry were strongly associated with the extent of AAC determined by CT scanning after an average of 10 years of follow-up [34]. However, this study did not report AGE measurements at the time of CT acquisition, so the relationship between AGE and AAC at the same time-point was not examined.

Secondly, we have also observed that both groups of patients with intermediate and high AGEs levels had a higher prevalence of diabetes-related complications. Several data have linked circulating AGEs to the development and progression of diabetes complications [35,36,37] including diabetic retinopathy [38,39,40,41], nephropathy [42,43], and cardiovascular disease [44,45,46]. Furthermore, an accumulation of AGEs has been detected in most of target tissues of diabetes complications, such as the kidney, retina, and atherosclerotic plaques [47]. Several studies have shown correlations between serum levels of AGEs and the development/severity of vascular disease [48,49,50,51,52]. Our results indicate that circulating levels of AGEs are mainly correlated with CKD and aging. These results are consistent with previous studies which found that serum AGEs are positively correlated with serum creatinine and inversely correlated with GFR [23,24,35,42,53]. Regarding aging and circulating levels of AGEs, our results are in accordance with other authors who indicated that AGEs are linked to the aging process [35,36,47,51,54,55] and are considered a marker of senescence [56].

On the other hand, we find an association between circulating AGEs and HbA1c. Similar results have been found in other studies [35] where AGEs levels were higher in patients with poor glycemic control measured by HbA1c [54,57,58,59]. It is important to consider that the extent and duration of hyperglycemia are predicted by increased levels of HbA1c, which is considered an acceptable marker. As mentioned above, prolonged exposure to glucose produces early AGEs and affects different proteins. An important example of early glycated proteins is HbA1c, which is further modified through a series of reactions in Hb-AGE. Under normal conditions, Hb-AGE makes up 0.42% of circulating hemoglobin (Hb) levels, increasing to 0.75% in diabetic subjects [60]. Besides HbA1c correlates with AGEs, AGEs-modified proteins are hardly degraded and remain in the vessels, kidney and heart for a long time, even after glycemic control has improved. Therefore, AGEs are considered to be a better index of cumulative diabetic exposure and one of the main actors of the metabolic memory observed in diabetic patients.

Concerning the factors associated to AAC, our results show that moderate-severe AAC is associated to current or ex tobacco use. Similar results were found in other studies, where cigarette smoking was strongly associated with calcification [61]. The independent factors, however, associated with AAC in our patients were age, male gender, AGEs levels and plasma triglycerides. These results are in accordance with previous studies which have reported that age and prior CV disease are strongly associated with AAC [62,63,64]. Triglycerides can be markers of glycemic control and have become an independent predictor of cardiovascular risk, as demonstrated by multiple prospective epidemiological studies [65,66,67,68]. Also, it is well-known that vascular calcification is more common in males than in females [69]. It is worth underlining that we developed ROC curves and searched for the optimal cut-off values associated with AAC. Remarkably, the accuracy for AGEs was higher than the observed for the other factors classically related to AAC (Figure 4). Moreover, an AUC higher than 0.8 showed a good fit for the multivariate model of factors associated to moderate-severe AAC (Figure 4). All these data suggest that AGEs could be useful markers of AAC which could identify those priority patients of diagnostic imaging techniques. The presence of calcification in any arterial wall is associated with a 3- to 4-fold higher risk for mortality and cardiovascular events [4]. Consequently, the early detection of AAC could have clinical importance in order to intensify cardiovascular prevention efforts. Therefore, it would be worth developing further prospective studies to draw definitive conclusions about the application of AGEs in algorithms for AAC prediction.

Our results support the notion that AGEs can play a role in the pathogenesis of aortic calcifications and are in accordance with tissue culture studies that indicate that AGEs can act as an inducer of calcification affecting cell function via both, intracellular glycation of proteins and AGE-RAGE signaling [20,22,70].

Considering that morbidity and mortality in patients with DM2 are mainly due to cardiovascular disease, the measurement of circulating AGEs could be useful as an early marker of the presence and severity of AAC and the occurrence of future diabetic complications. In addition, an inhibitory therapy of the formation of AGEs or a therapy addressed against the AGE-RAGE system could be useful as a preventive treatment in the development of AAC and diabetic complications Recently, DNA aptamers directed against AGE and RAGE, attenuating their interaction, have been developed and are being studied with promising experimental results [71]. Also, phytate has been described as inhibiting glycation processes by the iron chelation process [72] and is a well-known inhibitor of AAC via absorption over the hydroxyapatite crystals. Consequently, it is possible that this compound can be a strong inhibitor of the formation of AAC via the two aforementioned mechanisms [73,74,75]. Interestingly, a recent study suggests that the antidiabetic therapy with a sodium-glucose cotransporter inhibitor could reduce the levels of AGEs in comparison with a dipeptidyl peptidase−4 inhibitor [76]. Finally, beside drugs or compounds, the avoidance of smoking and the restriction of dietary AGEs consumption must be borne in mind to reduce the AGEs burden [77,78].

Our study has several limitations. First, the sample is small and it is a cross-sectional study from a single medical center; therefore, the findings presented here should be interpreted with caution. Even though we found that age, gender, triglycerides and circulating levels of AGEs were associated to AAC, this does not prove causality. Prospective multicentric longitudinal studies are needed to fully clarify how AGEs actually contribute to the genesis and progression of AAC and to assess the possibility that circulating levels of AGEs may serve as biomarkers to predict or diagnose AAC. Another possible limitation is that we measured AAC using lateral abdominal plain X-rays. One could argue that CT is more sensitive. Several studies, however, have positioned lateral lumbar radiography as a tool to estimate the severity of AAC. Furthermore, AAC evaluated by lateral X-rays shows a good correlation with coronary calcium score [4,79]. Therefore, we consider that plain radiography is an appropriate screening method for evaluating AAC in daily practice since it is simpler, cheaper and with lesser radiation than CT.

Finally, there are several methods available to measure AGEs and there are no single standardized protocols or normal reference ranges established for them. We measured a pool of AGEs by ELISA. It has been observed that CEL but not methylglyoxal hydroimidazolone, 3-deoxyglucosone hydroimidazolone, 2-aminoadipic acid and methionine sulfoxide was related to AAC [34]. Therefore, it is possible that our results were related to specific AGEs. Thus, further studies aimed at addressing these issues are needed.

## 5. Conclusions

In conclusion, we provide evidence that serum levels of AGEs positively correlate with AAC in patients with DM2. In addition, our study suggests that age, gender, triglycerides and circulating levels of AGEs are the risk factors associated to moderate–severe AAC. We also have shown that circulating levels of AGEs were positively associated with AAC severity and diabetes-related complications. Therefore, increased levels of AGEs can be contemplated as a biomarker of AAC and other vascular complications of diabetes.

## Figures and Tables

**Figure 1 jcm-09-01751-f001:**
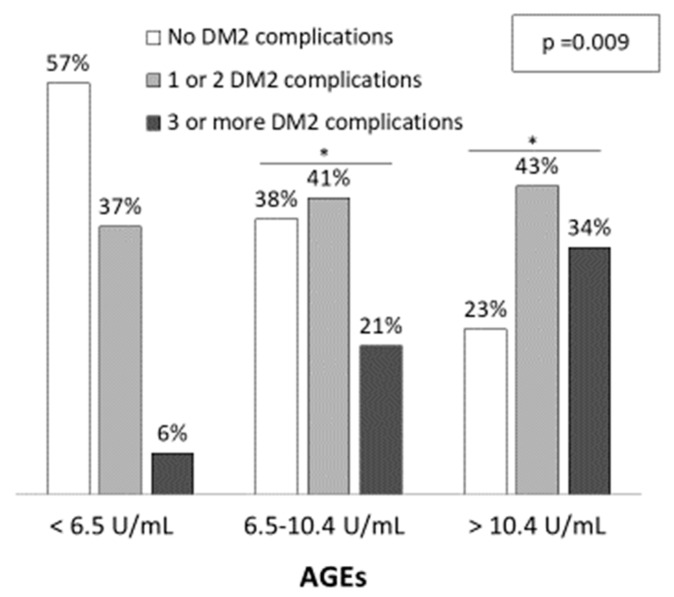
Diabetic related-complications among serum AGEs levels. The significance of differences between groups were determined using chi-square test (*p* = 0.009). * *p* < 0.05 vs. Low AGEs group (<6.5 U/mL).

**Figure 2 jcm-09-01751-f002:**
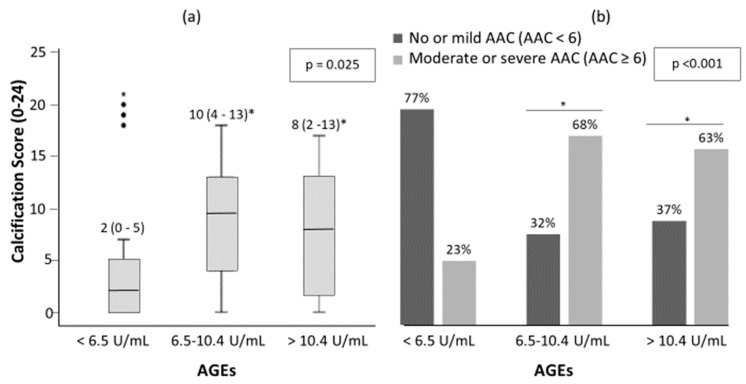
Abdominal aortic calcification (AAC) score (**a**) and percentage of patient with no-mild AAC and moderate-severe AAC (**b**) among AGEs levels in patients with DM2. The significance of differences between groups were determined using Kruskal–Wallis and Mann–Whitney U test for (**a**); and chi-square test for (**b**). * *p* < 0.05 vs. low serum levels of AGEs (<6.5 U/mL).

**Figure 3 jcm-09-01751-f003:**
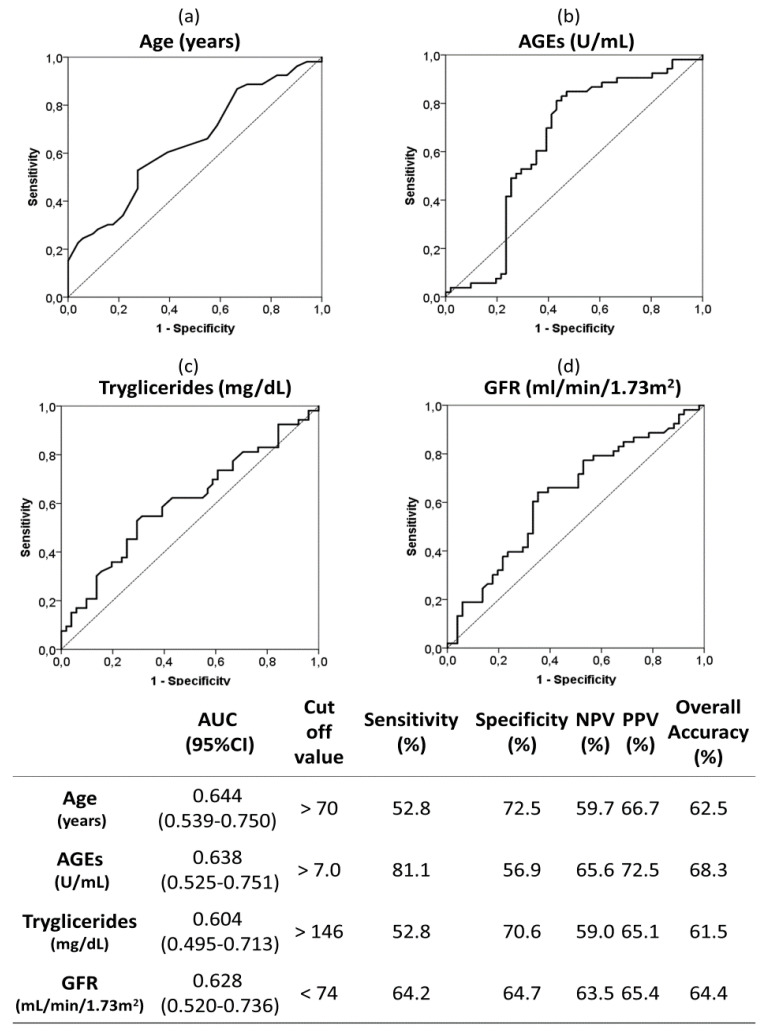
Receiver operating characteristic (ROC) curves and optimal cut-off of the following quantitative risk factors associated to moderate-severe AAC (vs. no-mild AAC): age (**a**), AGEs (**b**), triglycerides (**c**) and GFR (**d**). The optimal cutoff values were determined by the maximum Youden index (J), defined as sensitivity + specificity −1. The table indicate the area under the curve (AUC), sensitivity, specificity, positive predictive value (PPV), negative predictive value (NPV) and overall accuracy of the optimal cut off values. Abbreviations. AGEs; advanced glycation end products; GFR: glomerular filtration rate.

**Figure 4 jcm-09-01751-f004:**
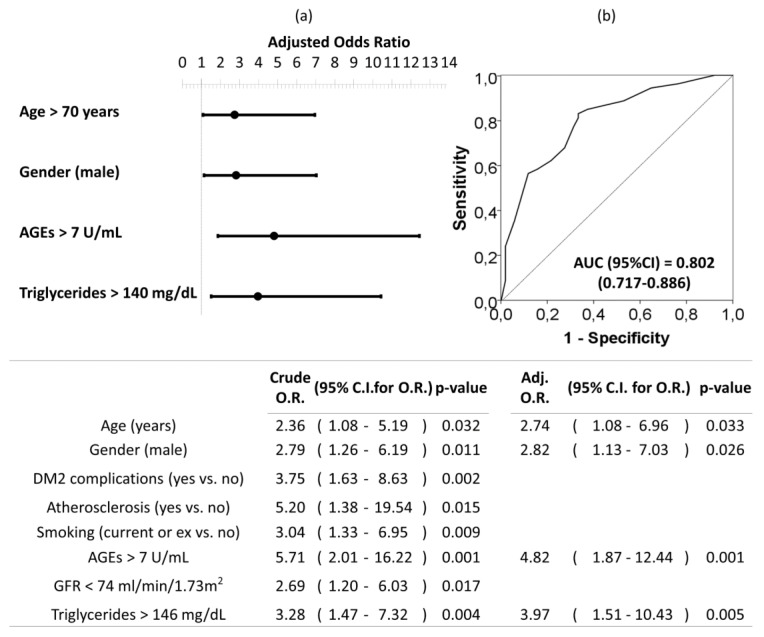
Forest plot (**a**) and ROC curve (**b**) of the multivariate logistic regression of risk factors associated to moderate–severe AAC in patients with DM2. Multivariate analysis was performed using stepwise backward method. Crude and adjusted odds ratio (OR) are indicated in the table. Comparison of the expected and observed frequencies by the Hosmer–Lemeshow goodness-of-fit test (*p*-value = 0.734) and by ROC curve (AUC = 0.802; *p* < 0.001) indicated a good fit for the model. AUC = the area under the curve, GFR = glomerular filtration rate.

**Table 1 jcm-09-01751-t001:** Clinical characteristics of patients among circulating advanced glycation end products (AGEs) groups.

	Low AGEs	Intermediate AGEs	High AGEs	*p*-Value
	(< 6.5 U/mL)	(6.5–10.4 U/mL)	(> 10.4 U/mL)	
	(*n* = 35)	(*n* = 34)	(*n* = 35)	
Age (years)	67.5 ± 7.3	66.9 ± 6.9	73.7 ± 7.9 ^a,b^	< 0.001
Gender (male)	12 (34.3%)	17 (50.0%)	21 (60.0%)	0.095
Body Mass Index (kg/m^2^)	31.6 ± 4.8	31.1 ± 4.4	31.4 ± 8.1	0.689
Waist circumference (cm)	110 ± 10	110 ± 11	108 ± 12	0.466
Systolic blood pressure (mmHg)	145 ± 17	145 ± 21	147 ± 19	0.827
Diastolic blood pressure (mmHg)	77.4 ± 10.7	76.5 ± 9.8	75.1 ± 12	0.827
Pulse pressure (mmHg)	67.5 ± 16.1	68.8 ± 17.5	72.3 ± 15.8	0.533
Heart rate (pulse/min)	78.1 ± 12.6	77.9 ± 13.7	76.4 ± 14.2	0.830
Time from diagnosis DM2				
Less than 5 years	4 (11.4%)	2 (5.9%)	3 (8.6%)	0.726
Between 5 and 10 years	10 (28.6%)	11 (32.4%)	7 (20.0%)	
More than 5 years	21 (60.0%)	21 (61.8%)	25 (71.4%)	
Comorbidities, toxics and clinical history				
Hypertension	31 (88.6%)	30 (88.2%)	32 (91.4%)	0.893
Smoking	9 (25.7%)	17 (50.0%)	14 (40.0%)	0.114
Alcohol	5 (14.3%)	5 (14.7%)	3 (8.6%)	0.688
Prior cardiovascular disease	15 (42.9%)	13 (38.2%)	20 (57.1%)	0.258
Chronic kidney disease (CKD)	27 (77.1%)	19 (55.9%)	32 (91.4%) ^b^	0.003
2 (GFR 89–60 mL/min/1.73 m^2^)	17 (48.6%)	9 (26.5%)	17 (48.6%)	0.100
3a (GFR 59–45 mL/min/1.73 m^2^)	5 (14.3%)	5 (14.7%)	8 (22.9%)	0.566
3b (GFR 44–30 mL/min/1.73 m^2^)	5 (14.3%)	5 (14.7%)	7 (20.0%)	0.772
Diabetic retinopathy	9 (25.7%)	13 (38.2%)	14 (40.0%)	0.393
Ischemic heart disease	4 (11.4%)	5 (14.7%)	9 (25.7%)	0.255
Cerebral stroke	3 (8.6%)	3 (8.8%)	8 (22.9%)	0.135
Diabetic polyneuropathy	7 (20.0%)	6 (17.6%)	11 (31.4%)	0.135
Diabetic foot	1 (2.9%)	1 (2.9%)	3 (8.6%)	0.442
Intermittent claudication	2 (5.7%)	2 (5.9%)	6 (17.1%)	0.207
Atherosclerosis	3 (8.6%)	2 (5.9%)	11 (31.4%) ^a,b^	0.007
Medication use				
Angiotensin-converting enzyme inhibitors/Angiotensin II receptor-blocking agents	22 (62.9%)	29 (85.3%)	24 (68.6%)	0.190
Beta-blockers	13 (37.1%)	9 (26.5%)	14 (40.0%)	0.462
Calcium antagonists	12 (34.3%)	9 (26.5%)	20 (57.1%)^b^	0.025
Statins	21 (60.0%)	29 (85.3%)	24 (68.6%)	0.062
Fibrates	2 (5.7%)	1 (2.9%)	2 (5.7%)	0.825
Phosphate binders	0 (0.0%)	1 (2.9%)	0 (0.0%)	0.354
25-hydroxyvitamin D	5 (14.3%)	6 (17.6%)	11 (31.4%)	0.178
Calcitriol or alphacalcidiol	1 (2.9%)	1 (2.9%)	0 (0.0%)	0.596
Paricalcitol	2 (5.7%)	0 (0.0%)	1 (2.9%)	0.366
Calcium supplement	4 (11.4%)	3 (8.8%)	3 (8.6%)	0.904
Bisphosphonates	3 (8.6%)	0 (0.0%)	2 (5.7%)	0.239
Steroids	0 (0.0%)	2 (5.9%)	0 (0.0%)	0.123
Antiplatelets	13 (37.1%)	27 (79.4%) ^a^	22 (62.9%) ^a^	0.001
Oral anticoagulants	4 (11.4%)	3 (8.8%)	5 (14.3%)	0.777
Insulin	19 (54.3%)	17 (50.0%)	20 (57.1%)	0.836
Oral antidiabetic	30 (85.7%)	30 (88.2%)	26 (74.3%)	0.262
Uric acid medication (allopurinol or febuxostat)	5 (14.3%)	5 (14.7%)	8 (22.9%)	0.566
Potassium citrate	0 (0.0%)	0 (0.0%)	1 (2.9%)	0.370
Thiazides	10 (28.6%)	10 (29.4%)	12 (34.3%)	0.856
Furosemide or triamterene	8 (22.9%)	9 (26.5%)	11 (31.4%)	0.719
Spironolactone or eplerenone	1 (2.9%)	2 (5.9%)	3 (8.6%)	0.591

Each value is given as mean ± standard deviation or frequency (percentage). The significance of differences between groups were determined using one-way analysis of variance (ANOVA) and the independent-samples *t*-test; or Kruskal–Wallis test and Mann–Whitney U test for quantitative data. Chi-square test and Fisher’s exact test were used for qualitative data. a: *p* < 0.05 vs. corresponding value of Low AGEs group; b: *p* < 0.05 vs. corresponding value of Intermediate AGEs group. Abbreviations. DM2: type 2 diabetes mellitus; GFR: glomerular filtration rate calculated by MDRD-4 IDMS equation.

**Table 2 jcm-09-01751-t002:** Laboratory parameters among AGEs groups.

	Low AGEs	Intermediate AGEs	High AGEs	*P*-Value
	(<6.5 U/mL)	(6.5–10.4 U/mL)	(>10.4 U/mL)	
	(*n* = 35)	(*n* = 35)	(*n* = 35)	
HbA1c (%)	7.2 (6.0–8.1)	7.2 (6.8–7.8)	8.0 (7.0–8.5) ^a,b^	0.026
Leukocytes (×109/L)	7.9 (6.7–9.6)	7.5 (6.3–9.2)	7.5 (6.3–9.5)	0.675
Hemoglobin (g/dL)	13.4 (12.5–14.5)	13.2 (12.6–14.8)	13.0 (11.4–13.8) ^a,b^	0.039
Glucose (mg/dL)	152.0 (125.0–194.0)	148.5 (125.5–191.8)	138.0 (110.0–230.0)	0.847
Urea (mg/dL)	38.0 (30.0–52.0)	41.0 (33.8–48.3)	49.0 (37.0–66.0) ^a^	0.026
Creatinine (mg/dL)	0.8 (0.7–1.1)	0.9 (0.7–1.2)	1.1 (0.9–1.4) ^a,b^	0.018
Urate (mg/dL)	5.2 (4.5–6.2)	5.4 (4.2–6.3)	6.4 (4.8–7.4) ^a,b^	0.033
Sodium (mg/L)	140.8 (139.0–141.0)	140.0 (138.7–141.2)	140.0 (138.0–142.0)	0.847
Potassium (mg/L)	4.5 (4.2–4.7)	4.5 (4.3–4.9)	4.7 (4.1–5.0)	0.350
Chloride (mg/L)	104.7 (103.0–106.0)	106.0 (102.8–107.2)	104.8 (103.0–107.0)	0.517
Calcium (mg/dL)	9.4 (9.1–9.9)	9.5 (9.3–9.8)	9.4 (9.2–9.6)	0.714
Magnesium (mg/dL)	1.8 (1.7–1.9)	1.8 (1.7–2.0)	1.7 (1.7–1.9)	0.326
Phosphate (mg/dL)	3.5 (3.3–3.8)	3.6 (3.2–3.8)	3.3 (3.1–3.7)	0.274
Total cholesterol (mg/dL)	158.0 (138.0–180.0)	154.0 (137.8–177.8)	154.5 (141.0–179.0)	0.991
HDL cholesterol (mg/dL)	42.5 (36.0–48.0)	39.5 (31.0–46.8)	38.0 (32.0–53.0)	0.597
LDL cholesterol mg/dL)	93.0 (69.2–104.8)	89.8 (66.3–101.6)	84.9 (70.0–101.8)	0.874
Triglycerides (mg/dL)	121.0 (84.0–191.0)	143.5 (99.8–198.5)	132.0 (103.0–203.0)	0.354
Albumin (g/dL)	4.2 (4.0–4.4)	4.1 (4.0–4.3)	4.0 (3.8–4.2)	0.106
Alkaline phosphatase (u/L)	78.0 (57.0–91.0)	73.5 (58.5–93.3)	81.5 (69.0–93.0)	0.512
PTHi (pg/mL)	67.0 (48.9–99.6)	73.0 (48.9–100.3)	69.7 (51.8–100.3)	0.993
25-hydroxyvitamin D (ng/mL)	23.0 (17.0–37.2)	26.8 (16.0–44.0)	26.8 (17.9–33.3)	0.670
GFR (mL/min/1.73 m^2^)	77.8 (55.7–88.9)	83.1 (50.1–98.2)	65.4 (46.0–79.1) ^a,b^	0.040
Urinary creatinine (mg/dL)	70.5 (44.9–89.7)	68.7 (54.6–100.3)	54.5 (36.5–88.1)	0.115
Urinary Alb/creat ratio (mg/g)	9.5 (4.2–22.2)	11.2 (5.5–98.6)	24.5 (9.4–47.5) ^a^	0.017
Urinary albumin (mg/dL)	0.6 (0.5–3.3)	0.8 (0.5–12.7)	1.1 (0.5–7.7)	0.069

Each value is given as median (interquartile range). The significance of differences between groups were determined using one-way ANOVA and the independent-samples *t*-test; or Kruskal–Wallis test and Mann–Whitney U test for quantitative data. a: *p* < 0.05 vs. corresponding value of Low AGEs group; b: *p* <0.05 vs. corresponding value of Intermediate AGEs group. Abbreviations. HbA1c: glycated hemoglobin; HDL: high-density lipoprotein; LDL: low-density lipoprotein; PTHi: intact parathyroid hormone; GFR: glomerular filtration rate calculated by MDRD-4 IDMS equation; Alb: albumin; Creat: creatinine.

**Table 3 jcm-09-01751-t003:** Clinical characteristics and laboratory parameters between patients with no-mild AAC and moderate–severe AAC.

	No-Mild AAC	Moderate-Severe AAC	*p*-Value
	(AAC < 6)	(AAC ≥ 6)	
	(*n* = 51)	(*n* = 53)	
Age (years)	67.3 ± 7.4	71.4 ± 8.1	0.011
Sex (female)	33 (64.7%)	21 (39.6%)	0.012
Body Mass Index (kg/m^2^)	31.6 ± 4.8	31.0 ± 7.0	0.174
Waist circumference (cm)	111 ± 10	108 ± 12	0.067
Systolic blood pressure (mmHg)	145 ± 17	147 ± 20	0.815
Diastolic blood pressure (mmHg)	77 ± 10	76 ± 11	0.951
Pulse pressure (mmHg)	69 ± 16	70 ± 17	0.682
Heart rate (pulse/min)	79 ± 12	76 ± 14	0.401
Time from diagnosis of diabetes			0.254
Less than 5 years	6 (11.8%)	3 (5.7%)	
Between 5 and 10 years	16 (31.4%)	12 (22.6%)	
More than 5 years	29 (56.9%)	38 (71.7%)	
Diabetic complications (nº)			0.002
1	10 (19.6%)	16 (30.2%)	
2	9 (17.6%)	7 (13.2%)	
3 or more	4 (7.8%)	17 (32.1%)	
Comorbidities, toxics and clinical history			
Chronic kidney disease (stage 2 or upper)	34 (66.7%)	44 (86.3%)	0.071
Hypertension	45 (88.2%)	48 (94.1%)	0.758
Smoking (current or ex)	13 (25.5%)	27 (52.9%)	0.009
Alcohol (current or ex)	8 (15.7%)	5 (9.8%)	0.386
Prior cardiovascular disease	20 (39.2%)	28 (54.9%)	0.175
Diabetic retinopathy	15 (29.4%)	21 (39.6%)	0.308
Ischemic heart disease	5 (9.8%)	13 (24.5%)	0.069
Cerebral stroke	5 (9.8%)	9 (17.0%)	0.391
Diabetic polyneuropathy	9 (17.6%)	15 (28.3%)	0.247
Diabetic foot	1 (2.0%)	4 (7.5%)	0.363
Intermittent claudication	2 (3.9%)	8 (15.1%)	0.930
Atherosclerosis	3 (5.9%)	13 (24.5%)	0.013
Laboratory parameters			
AGEs (U/mL)	5.9 (4.5–10.1)	9.7 (7.5–11.1)	0.015
HbA1c (%)	7.2 (6.4–8.1)	7.4 (6.9–8.4)	0.171
Leukocytes (×109/L)	7.8 (6.7–9.3)	7.5 (6.3–9.3)	0.509
Hemoglobin (g/dL)	13.4 (12.5–14.5)	13.0 (11.6–14.5)	0.164
Glucose (mg/dL)	152 (125–180)	139 (111–226)	0.701
Creatinine (mg/dL)	0.8 (0.7–1.1)	1.1 (0.8–1.4)	0.004
Urate (mg/dL)	5.2 (4.5–6.0)	6.2 (4.3–7.3)	0.036
Sodium (mg/L)	140 (139–141)	140 (138–142)	0.536
Potassium (mg/L)	4.4 (4.2–4.7)	4.7 (4.3–5.0)	0.022
Calcium (mg/dL)	9.5 (9.2–9.8)	9.4 (9.2–9.7)	0.577
Phosphate (mg/dL)	3.5 (3.2–3.8)	3.4 (3.1–3.8)	0.320
Total cholesterol (mg/dL)	158 (134–180)	154 (141–177)	0.964
HDL cholesterol (mg/dL)	42.5 (36.0–49.0)	38.0 (32.0–47.5)	0.199
LDL cholesterol mg/dL)	93.0 (68.3–105.2)	85.0 (67.8–100.9)	0.599
Triglycerides (mg/dL)	121 (94–176)	149 (103–203)	0.048
Albumin (g/dL)	4.2 (4.0–4.4)	4.0 (3.8–4.2)	0.004
Alkaline phosphatase (u/L)	77 (57–91)	82 (67–93)	0.356
PTHi (pg/mL)	67.0 (48.9–94.6)	74.2 (51.8–102.9)	0.333
25-hydroxyvitamin D (ng/mL)	23.0 (16.3–41.0)	26.8 (18.0–34.0)	0.547
GFR (mL/min/sup)	79.2 (57.4–95.3)	68.0 (47.1–82.3)	0.024
Urinary creatinine (mg/dL)	70.5 (52.4–89.7)	60.3 (39.2–89.7)	0.184
Urinary Alb/creat ratio (mg/g)	10 (5–29)	25 (9–149)	0.004
Urinary albumin (mg/dL)	0.6 (0.5–3.6)	1.1 (0.5–17.1)	0.012

Each value is given as mean (± standard deviation), frequency (percentage) or median (interquartile range). The significance of differences between groups were determined using independent *t*-test or Mann–Whitney U test for quantitative data and chi-square test or Fisher’s exact test for qualitative data. Abbreviations. AGEs: advanced glycation end products; HbA1c: glycated hemoglobin; HDL: high-density lipoprotein; LDL: low-density lipoprotein; PTHi: intact parathyroid hormone; GFR: glomerular filtration rate calculated by MDRD-4 IDMS equation; Alb: albumin; Creat: creatinine.

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
