# Peer review of "Role of Advanced Glycation End Products on Aortic Calcification in Patients with Type 2 Diabetes Mellitus"

_jcm, 2020, doi:10.3390/jcm9061751_

Round 1

Reviewer 1 Report

Thanks for your efforts for this study.

Review for Manuscript jcm-824320:

Title: Role of advanced glycation end products on aortic 3 calcification in patients with type 2 diabetes mellitus

Thank you for providing me with the opportunity to review the paper entitled "Role of advanced glycation end products on aortic 3 calcification in patients with type 2 diabetes mellitus".

General comments:

The manuscript investigates whether there is a correlation between serum levels of advanced glycation end products (AGEs) and abdominal aortic calcification (AAC) in patients with type 2 diabetes mellitus (DM2) and attempts to determine the most important risk factors of moderate-severe AAC in T2DM. The authors show that serum levels of AGEs positively correlate with AAC in 337 patients with DM2. In addition, the study provides evidence that circulating levels of AGEs are associated with AAC along with the other traditional factors related to AAC. The authors conclude that increased levels of AGEs may serve as a biomarker of AAC and other vascular complications of diabetes. Although this study has some interesting content, I believe the comments below need to be addressed:

Major comments:

  1. The authors analyzed the relation between circulating levels of AGEs and AAC severity with tertiles of serum AGEs levels and the median of AAC scores. Since both are continuous variables I suggest analyzing them to see if they have a linear correlation.
  2. No pairwise comparisons were made between the three groups of serum AGEs levels (low, intermediate, and high).
  3. The group of patients with high AGEs are significantly older than the patients in the other groups. Is there any chance that age is a confounding factor in the relationship of serum levels of AGEs and aortic complications and DM complications?
  4. As the authors state, a recent study suggests that the oral hypoglycemic sodium-glucose cotransporter inhibitor could reduce the levels of AGEs. Was there any difference in the prescription of sodium-glucose cotransporter inhibitors between the low, intermediate and high AGEs groups?
  5. The clinical implications of predicting AAC with circulating levels of AGEs should be highlighted.

Minor comments:

  1. “104.” on page 2, line 82 should not be boldfaced and the period should be deleted.
  2. The following corrections should be made:

1) Page 1 line 23: “aortic abdominal calcification AAC” → “abdominal aortic calcification (AAC)”

2) Page 2 line 71: they -> their

Author Response

We thank to the reviewer for his/her appreciation and constructive comments, which have led to a much improved version of our manuscript. All his/her comments/suggestions are addressed as detailed in this point-by-point reply and the changes in the manuscript have been highlighted.

Major comments:

Comment 1. The authors analyzed the relation between circulating levels of AGEs and AAC severity with tertiles of serum AGEs levels and the median of AAC scores. Since both are continuous variables I suggest analyzing them to see if they have a linear correlation.

Response to comment 1. We thank the reviewer for this remark and we would like to clarify that the reason why we decided not to use a linear correlation is because neither values of AGEs nor those of AAC score followed a normal distribution when normality plots and tests were applied. Nevertheless, the Spearman correlation coefficient between these two variables was 0.299 with a p-value of 0.003; and the Pearson correlation coefficient was 0.246 with a p-values of 0.014. These correlation coefficients are not really high because of the AAC score increase significantly from the low to intermediate AGEs tertile (Median [IQR]: 2[0-5] to 10 [4-13], respectively) but it didn’t increase from the intermediate to high AGEs tertiles groups (Median [IQR]: 10[4-13] to 8 [2-13], respectively) (figure 2).

Comment 2. No pairwise comparisons were made between the three groups of serum AGEs levels (low, intermediate, and high).

Response to comment 2. We thank the reviewer for this comment. We have included pairwise comparisons in tables 1 and 2. As can be seen in the caption of these tables, the significance of differences between AGEs groups were determined using one-way ANOVA and the independent-samples t-test (as a post-hoc test); or Kruskal-Wallis test and U Mann-Whitney U test (as a post-hoc test) for quantitative data. Chi-square test and Fisher’s exact test were used for qualitative data. All changes have been highlighted.

Comment 3. The group of patients with high AGEs are significantly older than the patients in the other groups. Is there any chance that age is a confounding factor in the relationship of serum levels of AGEs and aortic complications and DM complications?

Response to comment 3. We thank the reviewer for this comment. It is completely true that age is highly correlated with AGEs and AAC and it could be a confounding factor. Nevertheless, when multivariate binary logistic regression was performed (figure 4), AGEs levels were still statistically significant after adjusting with age>70 years, tryglicerides >146 mg/dL and gender (male). Furthermore, the crude and age-adjusted O.R. for AGEs >7 U/mL are shown in the next table (independent variable moderate-severe AAC):

Crude O.R. (95%CI)

O.R. adjusted by age as a continuous variable  (95%CI)

O.R. adjusted by age as dichotomized variable (> 70 years) (95 %CI)

AGEs > 7 U/mL

5.71 (2.01-16.22)

5.24 (2.13-12.90)

4.838 (1.99-11.77)

As can be seen, AGEs levels are significantly (p ≤ 0.001) associated to moderate-severe AAC after adjusting by age (as a continuous and categorical variable).

 Comment 4. As the authors state, a recent study suggests that the oral hypoglycemic sodium-glucose cotransporter inhibitor could reduce the levels of AGEs. Was there any difference in the prescription of sodium-glucose cotransporter inhibitors between the low, intermediate and high AGEs groups?

Response to comment 4. We thank the reviewer for this comment. Unfortunately, we didn’t register the sodium-glucose cotransporter inhibitors in the data base when this study was made. We only registered if patients were taking oral antidiabetic or not. Further studies should be developed in order to elucidate the effect of the oral hypoglycemic sodium-glucose cotransporter inhibitor in reducing the levels of AGEs.

Comment 5. The clinical implications of predicting AAC with circulating levels of AGEs should be highlighted.

Response to comment 5. We thank the reviewer for this comment which help to improve the manuscript. We have added a paragraph in the discussion section about the clinical implications of predicting AAC with circulating levels of AGEs. The changes have been highlighted.

Minor comments:

Comment 1. “104.” on page 2, line 82 should not be boldfaced and the period should be deleted.

Response to comment 1. We thank the reviewer for this comment which help to improve the manuscript. We have made these corrections in the new version of the manuscript and the changes have been highlighted.

Comment 2. The following corrections should be made:

1) Page 1 line 23: “aortic abdominal calcification AAC” → “abdominal aortic calcification (AAC)”

2) Page 2 line 71: they -> their

Response to comment 2. We thank the reviewer for this comment which help to improve the manuscript. We have made these corrections in the new version of the manuscript and the changes have been highlighted.

Reviewer 2 Report

Τhis is a very interesting paper. The subject is emerging and useful.  The Abstract is sufficient. The Introduction is accurate. The experimental design is flawless. The analysis is robust. The results are clear. The Discussion is fulent and elegant with due interpretations. The literature is to the point and well chosen. In the opinion of this reviewer, the above mentioned paper is a very useful contribution.

This is a well-written paper that does not need corrections

Author Response

We thank to the reviewer for his/her appreciation and constructive comments.

Round 2

Reviewer 1 Report

Thanks for your efforts to revise the manuscript.